# Noise robustness of persistent homology on greyscale images, across filtrations and signatures

**Renata Turkeš** [1]*, **Jannes Nys**[1], **Tim Verdonck** [2], **Steven Latré**[1]

**1** Department of Computer Science, IDLab, University of Antwerp - imec, Antwerp, Belgium, **2** Department of Mathematics, Applied Mathematics, University of Antwerp, Antwerp, Belgium

* renata.turkes@uantwerpen.be

**Data Availability Statement:** All relevant data are available at http://yann.lecun.com/exdb/mnist/. The code is available at https://github.com/renata-turkes/turkevs2021noise. The code allows other researchers to replicate our study, or to easily investigate the noise robustness of PH with

## Abstract

Topological data analysis is a recent and fast growing field that approaches the analysis of datasets using techniques from (algebraic) topology. Its main tool, persistent homology (PH), has seen a notable increase in applications in the last decade. Often cited as the most favourable property of PH and the main reason for practical success are the stability theorems that give theoretical results about noise robustness, since real data is typically contaminated with noise or measurement errors. However, little attention has been paid to what these stability theorems mean in practice. To gain some insight into this question, we evaluate the noise robustness of PH on the MNIST dataset of greyscale images. More precisely, we investigate to what extent PH changes under typical forms of image noise, and quantify the loss of performance in classifying the MNIST handwritten digits when noise is added to the data. The results show that the sensitivity to noise of PH is influenced by the choice of filtrations and persistence signatures (respectively the input and output of PH), and in particular, that PH features are often not robust to noise in a classification task.

## Introduction

Homology goes back to the beginnings of topology in Poincaré's influential papers, who represented the notion of a connectivity of a space with its cycles of different dimensions (e.g., 0-, 1-, and 2-dimensional cycles respectively correspond to connected components, loops, and cavities). These cycles are shown to organize themselves into abelian groups, called homology groups, and their ranks (referred to as the Betti numbers of the space) are non-negative integers corresponding to the number of independent cycles in each dimension [1]. This homology information can be very useful, as it allows to classify spaces and uncover the underlying structure of a space. For a detailed study of homology, we refer to [2].

Real data are a finite set of observations and do not directly reveal any topological information, since topological features are usually associated with continuous spaces. To circumvent this issue, the underlying topological structure of the data (e.g., a point cloud, a finite set of data points in space) can be estimated at different scales with a nested family of topological

different filtrations and persistence signatures, and their parameters and corresponding metrics, for other datasets of greyscale images, types of noise and/or classifiers.

**Funding:** The author(s) received no specific funding for this work.

**Competing interests:** The authors have declared that no competing interests exist.

spaces, called filtration. The filtration is used to calculate the information about $k$-dimensional cycles that *persist* across different scales of data, referred to as persistent homology (PH) [3–5]. More precisely, $k$-dimensional PH registers the scale (also referred to as resolution, or time) at which every $k$-dimensional cycle appears and disappears in the filtration. This PH information can be represented using different signatures, e.g., using sets, vectors, functions, or scalars. The pipeline for PH is visualized in Fig 1, and explained in greater detail in the next section. For a gentle, but detailed introduction to PH for a broad range of computational scientists, see [6].

Over the past two decades, persistent homology has found many applications in data science, e.g., in the analysis of local behaviour of the space of natural images [7], analysis of images of hepatic lesions [8], human and monkey fibrin [9], fingerprints [10], or diabetic retinopathy images [11], analysis of 3D shapes [12, 13], neuronal morphology [14], brain artery trees [15, 16], fMRI data [17–19], protein binding [20], genomic data [21] orthodontic data [22], coverage in sensor networks [23], plant morphology [24], fluid dynamics [25], dynamical systems describing the movement of biological aggregations [26], cell motion [27], models of biological experiments [28], force networks in granular media [29], structure of amorphous and nanoporous materials [30, 31], spatial structure of the locus of afferent neuron terminals in crickets [32], or spread of the Zika virus [33]. An exhaustive collection of applications of topological data analysis to real data can be found at [34].

The main reason behind the recent popularity of persistent homology in data analysis is its proven stability: PH is robust under small perturbations in the input, which is of crucial importance for practical applications due to the unavoidable presence of noise or measurement error in real data [35]. Moreover, PH is commonly assumed to be a topological invariant and therefore robust under affine transformations.

However, it is often overlooked in the literature how strongly the stability theorems are influenced by the choice of a:

- filtration, the input for PH, or the medium through which the homology information is extracted from data. Indeed, it is important to remember that PH is not directly calculated on the data (e.g., an image, or a point cloud, see the first column in Fig 1), but on the filtration that approximates the shape of data at different scales (see the second column in Fig 1). The filtration must satisfy the underlying assumptions in the stability theorem, which then ensures robustness under minor perturbations in the input—filtration, not necessarily under minor perturbations of the data. Moreover, the level of robustness is directly determined by the filtration.

- persistence signature, the output of PH, or the medium used to represent PH. Indeed, the stability theorems do not provide a guarantee of the noise robustness of PH in general, but rather prove the stability of a selected signature (with the corresponding metric) (particular choice in the third or fourth column in Fig 1).

In addition, the choice of filtration influences the type of information captured with PH: for some filtrations, PH can reveal geometric information and thus not be invariant e.g., under rotation or translation. Furthermore, even if the stability theorem holds for the given filtration and signature, little attention has been paid to what these stability theorems mean in practice. In particular, it is unclear if the stability results imply the noise robustness of PH features in a classification task.

To investigate these issues, we carry out computational experiments that evaluate the noise robustness of PH on the MNIST dataset of greyscale images, under different types of noise. More precisely, the main objective of this work is to address the following research questions, across different filtrations and persistence signatures:

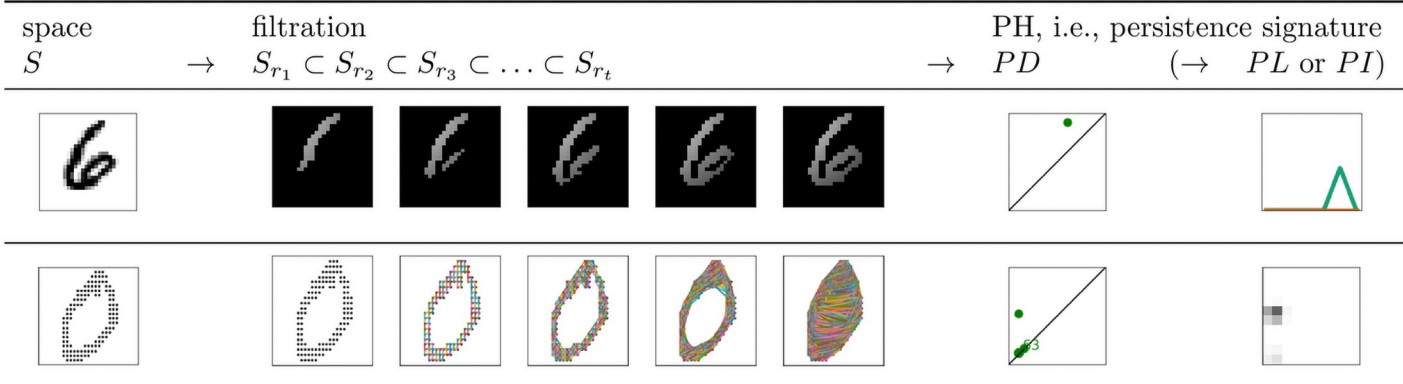

**Fig 1. Persistent homology pipeline.** PH can be calculated for different types of spaces *S*, which can represent a single data observation (typical for classification tasks) or a complete dataset. In this figure, we calculate the PH information for an image. The input for PH is a filtration, a nested family of spaces that approximate the structure of *S* at different scales $r_1 < r_2 < \ldots r_t$. For example, to approximate the structure of an image at scale *r*, we can look only at pixels within distance *r* from the top left pixel (top panel). Alternatively, we can look at an image as a point cloud, and approximate its structure at resolution *r* by constructing an edge between two points whenever they are within distance *r* (bottom panel). For a homological dimension *k* (in the figure, *k* = 1), PH registers the birth and death time *r* of every *k*-dimensional cycle (connected component, loop, void, etc.) within the filtration, and is commonly summarized with a scatter plot of birth and death coordinates, referred to as persistence diagram (*PD*). It is often interesting or even necessary to transform the *PD* into a different persistence signature, such as a persistence landscape (*PL*, top panel) or a persistence image (*PI*, bottom panel).

(RQ1)   How much does PH change under noise in the data?

(RQ2)   How discriminative is PH if the data contains noise?

The findings of this paper can therefore help to guide the choice of appropriate filtrations and signatures, especially in the presence of noise in the data. To the best of our knowledge, this issue has not been studied in literature so far. In the majority of studies that apply PH to tackle a particular problem (and are thus not concerned with noise robustness in particular), a single filtration and signature are commonly adopted, without a discussion on the motivation, assumptions, and implications behind the specific choice. There are a few noteworthy examples in the literature, such as [36], which do consider multiple filtrations and/or signatures (on the MNIST dataset), but they focus on the discriminative power, rather than the noise robustness of PH features. The authors do conclude, however, that PH is reputed for its robustness to noise, and suggest to conduct a similar study under different types of image noise [36].

The next section introduces the filtrations, persistence signatures and stability theorems. We then proceed to evaluate the robustness of PH on the MNIST image dataset of handwritten digits. The final section summarizes the findings and limitations of this work, and provides suggestions for future research.

## Materials and methods

This section provides more details about filtrations and persistence signatures, the input and output of persistent homology, and the stability theorems. We focus on a few common examples of filtrations and signatures that will be used in our computational experiments, and also discuss our choice of parameters.

### Filtrations

Persistent homology can be calculated for various types of space *S*, whether it represents point cloud, time series, graph or image data. To extract the PH information from a space, one must define a suitable filtration. The construction of a filtration in general relies on structured

complexes, a type of topological space that is particularly important in algebraic topology due to their combinatorial nature that allows for the computation of homology.

When the space is a point cloud $X \subset \mathbb{R}^n$, the most common choice for a structured complex is the simplicial complex, a set composed of simplices (points, line segments, triangles, tetrahedrons, and their $k$-dimensional counterparts, embedded in $\mathbb{R}^n$), that is closed under taking subsets (so that, for instance, if a triangle is in the simplicial complex, then all its edges and vertices are also elements of the simplicial complex) [5, 6]. Probably the most well-known is the Vietoris-Rips simplicial complex $VR(X, r)$ [37], built by constructing (i) a line segment for any pair of points in $X$ within distance $r$ of each other, (ii) a triangle, if the points in a triplet are all within distance $r$ of each other, and so forth. Different values of the so-called resolution parameter $r$ create different simplicial complexes and reveal different cycles. Hence, a single value of $r$ captures information about the space $X$ only at the given scale. However, the filtration $VR(X)$, defined as the nested family of subspaces

$$VR(X, r_1) \subseteq VR(X, r_2) \subseteq \cdots \subseteq VR(X, r_t),$$

can be used to depict how $k$-dimensional cycles persist across different values $r_1 < r_2 < \ldots r_t$ of the resolution $r$ (see Fig 1, bottom panel).

A *filtration* can alternatively be calculated using a *filtration function* $\phi : \mathbb{R}^n \to \mathbb{R}$, simply by considering the sublevel sets of $\phi$, determined by a scale cut-off $r$:

$$X_r = \{\mathbf{y} \in \mathbb{R}^n \mid \phi(\mathbf{y}) \leq r\} \quad (r \in \mathbb{R}).$$

The Rips filtration is obtained with $\phi = \delta_X$, where $\delta_X(\mathbf{y})$ is the minimum distance between $\mathbf{y} \in \mathbb{R}^n$ and any point $\mathbf{x} \in X$ on the point cloud. Indeed, according to the definition of the sublevel set of the distance function $\delta_X$,

$$X_r = \{\mathbf{y} \in \mathbb{R}^n \mid \delta_X(\mathbf{y}) \leq r\} = \cup_{x \in X} B(\mathbf{x}, r),$$

where $B(\mathbf{x}, r)$ is a ball with radius $r$ centred around $\mathbf{x}$; this union of balls approximates $VR(X, r)$ [35, 38]. However, the distance function $\delta_X$ is extremely sensitive to outliers and noise ("even one outlier is deadly", or, in the language of robust statistics, the distance function has breakdown point zero [39]). To circumvent this issue, [38] propose to rather consider distance-to-a-measure (DTM) $\delta_{X,m}$ as the filtration function, which is defined as the average distance from a given number of nearest neighbours in $X$ (and is thus a smooth version of the distance function) [40]. The number of neighbours that are considered is determined by the parameter $m$, which represents the percentage of the total number of point cloud $X$ points. In our computational experiments, we will quantify how much robustness to noise is actually gained in practice with PH on the DTM (with $m = 0.1$, a commonly suggested and typically default value) compared to the Rips filtration.

In this paper, we are interested in calculating PH for an image. Let $Z$ be a greyscale image, i.e., $Z = [z_{uv}]$, where $z_{uv}$ is the greyscale value of the pixel $(u, v)$, $u \in \{1, 2, \ldots, n_x\}$, $v \in \{1, 2, \ldots, n_y\}$, $n_x$ and $n_y$ are the numbers of pixels in respectively $x$ and $y$ direction. We can consider the image $Z$ as a 2D point cloud $X(Z, z_0) \subset \mathbb{R}^2$ consisting of all $(u, v) \in \mathbb{R}^2$ corresponding to pixels with a greyscale value above a fixed user-given threshold $z_0$. A possibility is then to define the filtration for the image $Z$ via the Rips (Fig 1, bottom panel) or DTM filtration of the point cloud $X(Z, z_0)$.

However, a point cloud (and its simplicial complex) is not the most natural representation of an image. Indeed, this representation does not exploit the natural grid structure of images [36]. For an image $Z$, we can rather consider its so-called cubical complex $K(Z)$, the cubical analogue to a simplicial complex, in which the role of simplices is played by cubes of different

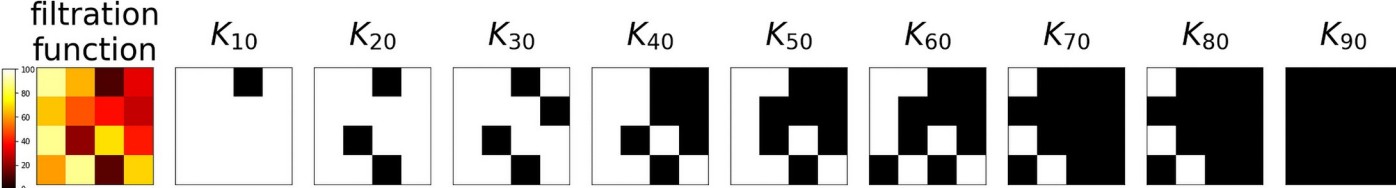

**Fig 2. Filtration on a cubical complex.** The first image represents the values [0, 100] of the filtration function $\phi$. The next nine figures show the cubical complexes $K_{10} \subseteq K_{20} \subseteq K_{30} \subseteq \cdots \subseteq K_{90}$, where $K_r$ corresponds to the union of all cubes, i.e., pixels $(u, v)$ with the filtration value $\phi(u, v) \leq r$. There is only one 1-dimensional cycle, i.e., hole (one-pixel hole in the third row and third column), which is first seen in $K_{40}$, and then disappears or closes in $K_{70}$.

dimension (points, line segments, squares, cubes, and their $k$-dimensional counterparts) [41]. The squares correspond to the image pixels, the edges to the sides of the pixels, and the points to the pixel corners. Another way to construct a cubical complex from an image is to consider the dual of the cubical complex defined above: the points reflect the pixels, the line segments the intersections of pairs of non-diagonally neighbouring pixels, and squares reflect the intersections of four pixels [42].

To define a filtration function $\phi : K(Z) \to \mathbb{R}$, it is thus necessary to define the value of $\phi$ on each cube in $K(Z)$. A natural filtration function on a cubical complex assigns to each square the value of the image on the corresponding pixel, and we use $\phi(u, v)$ to denote the value of the filtration function on the square corresponding to pixel $(u, v)$. The filtration function on the line segments and points is defined as the minimum value of all bordering pixels. A natural filtration function on the dual cubical complex assigns the pixel values as the values of the function on the points, and sets the function values for line segments and squares as the maximum value of all bordering simplices. These two methods differ with respect to the diagonally neighbouring pixels, as they are considered connected with the first approach, but not the second, what can result in substantially different persistent homology [42]. From such a filtration function, it is straightforward to build a filtration $K_{r_1} \subseteq K_{r_2} \subseteq \cdots \subseteq K_{r_t}$, where $K_r$ is the union of all cubes corresponding to pixels $(u, v)$ with $\phi(u, v) \leq r$ (Fig 1, top panel, and Fig 2). This family of nested subspaces is commonly referred to as the filtered cubical complex. Note that this means that (the cubes corresponding to) the pixels with the lowest filtration function value appear first and persist the longest in the filtration.

In this paper, we consider the following filtration functions $K(Z) \to \mathbb{R}$ on the cubical complex $K(Z)$ of an image $Z = [z_{uv}]$, described previously in [36].

- binary: The binary filtration function considers binary values of pixels by introducing a greyscale threshold $z_0$:

$$\phi_{z_0}(u, v) = \begin{cases} 0 & z_{uv} \geq z_0 \\ 1 & \text{otherwise.} \end{cases}$$

PH with respect to this filtration function corresponds to the homology of the image [36], meaning that it only determines the *number* of cycles (Betti numbers). It is of crucial importance that the greyscale threshold parameter $z_0$ is sufficiently low, so that all dark pixels are part of the filtration immediately at scale $r = 0$. Indeed, if only a single pixel along some hole has a greyscale value below the given threshold $z_0$, this pixel will only be a part of the filtration at resolution $r = 1$, as any other pixel in the image, so that the hole is never seen at any scale in the filtration.

- greyscale: In order to study how cycles persist with respect to the greyscale value, a nonbinary filtration function is a more natural choice. In the greyscale filtration function, one relates each pixel to its greyscale value:

$$\phi_{\text{grsc}}(u, v) = \max(Z) - z_{uv}.$$

An advantage of the greyscale compared to other considered filtrations is that it is parameter-free. In particular, it does not require an a-priori defined greyscale threshold. Next to the number of cycles, PH with respect to the greyscale filtration function thus also captures information about the brightness of the cycles.

- density: If the greyscale value of a single pixel changes significantly (e.g., from black to white), an existing hole in an image might get disconnected, or an additional single-pixel hole might appear. To avoid such sensitivity to outlying greyscale values, we can rather consider the density filtration function. Thereby, we relate each pixel to the number of "dark-enough" pixels in its neighbourhood. More precisely, let the neighbourhood $N((u, v), d_0, z_0)$ be the set of all pixels $(u', v')$ with $z_{u'v'} \geq z_0$ (for a given threshold $z_0$), that are within given distance from pixel $(u, v)$:

$$\|(u', v') - (u, v)\|_2 \leq d_0.$$

Density filtration function is then defined as:

$$\phi_{d_0, z_0}(u, v) = N(d_0) - |N((u, v), d_0, z_0)|,$$

where $N(d_0)$ is the total number of pixels within distance $d_0$, for any $(u, v)$. The threshold parameter $z_0$ is not of crucial importance. For instance, if only one pixel along a hole is very bright, the hole will never be seen in the binary filtration, but it will persist from early on in the density filtration, for most of the values of $z_0$. A good choice for the size of the neighbourhood $d_0$ obviously depends on the size of the image. For the dataset of $28 \times 28$ MNIST images, we take $d_0 = 1$.

- radial: While the greyscale and density filtration capture information about the brightness of cycles, it is possible to capture other information. For example, the *position* of cycles is captured with PH if one considers the radial filtration function defined as the distance from a given reference pixel $(u_0, v_0)$:

$$\phi_{(u_0, v_0), z_0}(u, v) = \begin{cases} \|(u, v) - (u_0, v_0)\|_2 & z_{uv} \geq z_0 \\ \max_{(u', v')} \|(u', v') - (u_0, v_0)\|_2, & \text{otherwise.} \end{cases}$$

Similar as for the binary filtration function, the greyscale threshold $z_0$ is crucial for the radial filtration as well, whereas the density and Rips filtration are less sensitive to this parameter (point cloud points corresponding to non-neighbouring pixels can still be connected with an edge, for a sufficiently large resolution $r$). However, to be consistent, we take the same threshold value $z_0 = 0.5 \max(Z)$ for the Rips, DTM, binary, density and radial filtrations. The choice of the reference pixel $(u_0, v_0)$ depends on where the important topological features are expected to be located in an image, and how this location differs across classes of data. For instance, if we consider $(u_0, v_0)$ to be a pixel in the centre of the image, the holes in digits 6 and 9 would be seen at the same resolution $r$ in the filtration. Since we aim to differentiate between digits 6 and 9, we will consider $(u_0, v_0) = (0, 0)$.

Fig 3 visualizes the filtration functions discussed in this section.

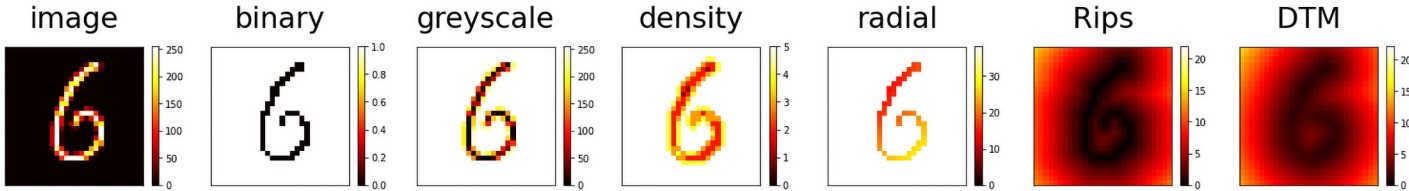

**Fig 3. Filtrations.** The first plot shows an example MNIST image $Z$, with greyscale pixel values in [0, 250]. The next four plots respectively show the heat map for the binary, greyscale, density and radial filtration function $\phi : K(Z) \to \mathbb{R}$, where $K(Z)$ is the cubical complex corresponding to the given example image. The final two plots visualize the heat map of $\phi : K(Z) \to \mathbb{R}$, where $\phi$ is the discretized version of the Rips and DTM filtration functions $\delta_{X(Z,z_0)} : \mathbb{R}^2 \to \mathbb{R}$ and $\delta_{X(Z,z_0),m} : \mathbb{R}^2 \to \mathbb{R}$, and $X(Z, z_0)$ is the point cloud corresponding to the image $Z$.

Persistent homology information in dimension $k$ captures the values of resolution $r$ when each $k$-dimensional cycle is born and when it dies in a filtration, denoted with $b$ and $d$. The cardinality of this multi-set of persistence intervals $(b_i, d_i)$ ($i \in \mathbb{N}^+$) counts the number of $k$-dimensional cycles (although many or even a majority might only show up in the filtration for a brief while, i.e., for a small range of resolution values $r$, yielding very short lifespans or persistence $l_i = d_i - b_i$, and as will be explained later in the paper, can thus be considered as irrelevant). However, the choice of the filtration defines the interpretation of the birth values and death values, which can reflect additional topological or geometric information, such as their size or position (Fig 4).

Obviously, the choice of filtration has an important influence on the noise robustness and discriminative power of PH. If PH only registers the *number* of holes, it is of topological nature and is invariant under rotations, translations, or stretching (in topology, a coffee mug is equivalent to a doughnut), which can be useful in some applications, such as recognition of animals, cars, or people in images. If PH also captures the position of holes, it is sensitive to rotation, but able to differentiate, e.g., between digits 6 and 9. If the size of the holes is also captured, the PH information is not robust to rescaling, but it enables us to differentiate between a 6 and a 0.

In this paper, we consider the **binary-, greyscale-, density-, and radial-filtered cubical complexes, and the Rips and DTM-filtered simplicial complexes** as the input for PH. Other filtrations have been introduced in the literature, such as the kernel distance or kernel density estimate (which inherit some reconstruction properties of DTM) [43], dilation (the smallest distance to a black pixel, thus representing a cubical analogue to the Rips filtration), erosion (inverse dilation), signed distance (a combination of dilation and erosion) [36], etc. Our goal is to emphasize how PH with different filtrations capture different information, and our selection is thus sufficient to illustrate this issue.

## Persistence signatures

As already mentioned, $k$-dimensional persistent homology is a multi-set of intervals $(b_i, d_i)$, with $b_i$ and $d_i$ corresponding to the scale $r \in \mathbb{R}$ when a $k$-dimensional cycle $i$ appears and disappears in the filtration. In order to represent this information visually, or to apply statistical inference or machine learning on PH, different methods are available, where the multi-set of birth-death values is represented using diagrams, functions, vectors, or even scalars. Below we provide an intuitive introduction to the persistence signatures used in this paper, which are the most common in the literature, and refer the reader to the relevant references for more details.

- **persistence diagram** (*PD*): Persistence diagram is the most straightforward representation of PH as a scatter plot of points $(b_i, d_i)$, counted with their multiplicity, and union all points on the diagonal, counted with infinite multiplicity [44, 45] (Fig 1).

| filtration | 1-dimensional PH | | | | | | | topological or geometric information about a hole |
|---|---|---|---|---|---|---|---|---|
| |  5 |  0 |  0 |  6 |  6 |  9 |  8 | |
| binary | ∅ | (0, 1) | (0, 1) | (0, 1) | ∅ | (0, 1) | (0, 1)<br>(0, 1) | - |
| greyscale | ∅ | (17, 255)<br>(159, 171) | (17, 255)<br>(159, 171) | (0, 255) | ∅ | (14, 255)<br>(213, 231) | (9, 255)<br>(19, 255)<br>(65, 86)<br>(223, 242)<br>(234, 255) | greyscale along and within hole |
| density | ∅ | (1, 5) | (1, 5) | (2, 5) | (3, 5) | (1, 5)<br>(3, 4)<br>(4, 5) | (1, 5)<br>(2, 5)<br>(3, 4) | density along and within hole |
| radial | ∅ | (25.00, 38.18) | (25.00, 38.18) | (24.76, 38.18) | ∅ | (20.00, 38.18) | (20.12, 38.18)<br>(24.60, 38.18) | position |
| Rips | (1.00, 1.41)* | (1.00, 1.41)*<br>(1.00, 8.94)<br>(2.00, 2.24) | (1.00, 1.41)*<br>(1.00, 6.40)<br>(2.00, 2.24)<br>(5.00, 6.00) | (1.00, 1.41)*<br>(1.00, 5.00) | (1.00, 1.41)*<br>(2.00, 5.00) | (1.00, 1.41)*<br>(1.00, 4.12)<br>(2.83, 3.00) | (1.00, 1.41)*<br>(1.00, 3.00)<br>(1.00, 3.60)<br>(2.83, 3.00) | sparsity, distance across hole |
| DTM | (4.13, 4.36)** | (4.40, 4.60)**<br>(4.94, 12.81) | (4.08, 4.38)**<br>(11.51, 11.95)<br>(4.94, 12.32) | (4.08, 4.33)**<br>(5.49, 8.60) | (4.39, 4.54)**<br>(6.10, 8.74) | (4.16, 4.58)**<br>(6.17, 6.19)<br>(4.16, 7.43) | (4.02, 4.33)**<br>(6.43, 6.51)**<br>(4.34, 6.57)<br>(5.00, 7.43) | sparsity, size |

**Fig 4. Persistent homology across filtrations.** Persistent homology is a multi-set of persistence intervals $(b_i, d_i)$, where $b_i$ and $d_i$ are respectively the time when a cycle $i$ (a connected component, loop, void, etc.) is born, and when it dies in a filtration. The table lists 1-dimensional PH calculated for a few example MNIST images (or an image with an outlying pixel), across selected filtrations. The notation $(b, d)^*$ implies that multiple cycles appear and disappear at the same time (thus, PH is a *multi*-set, where each element has its multiplicity). The notation $(b, d)^{**}$ implies that there are multiple intervals with a similar birth and death value. The cardinality of the set of persistence intervals determines the number of cycles. However, the definition of the filtration implies the interpretation of birth and death times, so that PH with different filtrations captures different topological (and geometric) information, what further influences its noise robustness and discriminating power. For example, an additional point at an outlying distance from a point cloud can have an important influence on PH with the Rips filtration (e.g., an additional black pixel within a hole will change the persistence of that hole, see persistence intervals in red), but this is less true for the DTM filtration, as the outlier will have a large distance from the nearest point cloud neighbours and will thus appear only very late in the filtration. A reverse example is a pixel with an outlying greyscale value (e.g., white pixel in a dark region) which has an important influence on PH with the binary, greyscale and radial filtration (in blue), but much less for the density, Rips and DTM filtration. If geometric information is captured, PH becomes sensitive under some affine transformations. Furthermore, 1-dimensional PH with binary, greyscale and density filtration cannot differentiate between digits 0, 6 and 9 (as they all have one hole of similar brightness), but radial filtration allows to discriminate between digits 6 and 9 (as the holes have a different position), and the Rips and DTM filtration enable to distinguish between 0 and 6 (as the holes are of different size).

An advantage of *PD*s compared to other persistence signatures is that they are parameter-free, but they also have an important disadvantage: they are not convenient for statistical inference, because their complicated structure makes common algebraic operations—such as addition, division, and multiplication—challenging [9] (so that, for instance, the mean might not be unique [46]). Furthermore, although *PD*s can be equipped with a metric structure (discussed below) which enables to perform a variety of machine learning techniques such as some clustering algorithms, many other machine learning tools (decision tree classification, neural networks, feature selection, some dimension reduction methods, and others) require more than a metric structure—they require a vector space [35].

- **(vectorized) persistence landscape** (*PL*): Persistence landscape is a function $\lambda : \mathbb{N} \times \mathbb{R} \to \mathbb{R}$ obtained by "stacking isosceles triangles" whose bases are the PH intervals $(b_i, d_i)$, and

whose heights reflect the so-called lifespans (or persistence) $l_i = d_i - b_i$ (Fig 1, top panel). Alternatively, it may be thought of as a sequence of functions $\lambda_j : \mathbb{R} \to \mathbb{R}$, where $\lambda_j(r) = \lambda(j, r)$ depicts how long the $j$-th most dominant cycle has lived until the moment $r$ in the filtration ($r - b_i$), or how long from $r$ before it dies ($d_i - r$) (Fig 1, top panel, two functions in green and orange).

In contrast to $PD$s, persistence landscapes lie in a Banach space, and are thus easy to combine with tools from statistics: they obey a strong law of large numbers and a central limit theorem, and the space of landscapes does have a unique mean [47]. However, for many machine learning tasks, it is necessary to consider finite vectors rather than functions, and a discretization of the function $\lambda$ into a vector $PL$ requires two additional parameters: we need to decide on the maximum number of first landscape functions $\lambda_j$ to consider, and on the number of points where each of these functions is evaluated, referred to as the landscape resolution. The number of main connected components or holes in the MNIST dataset is typically 0, 1 or 2. However, additional cycles might appear in noisy images, and we thus consider the first 10 landscapes ($j \in \{1, 2, \ldots, 10\}$); although we immediately note that this means that $PL$s and $PD$s do not necessarily capture the same information (e.g., if there are more than 10 cycles in this case). Obviously, this number should be higher if we expect a large number of important cycles that discriminate between classes of data. We set the landscape resolution to 100.

- **persistence image** ($PI$): Persistence image is constructed by superimposing a grid over a $PD$, and depicting the volume below the weighted sum of Gaussian probability density functions, on each grid cell [35] (Fig 1, bottom panel). This is a more sophisticated variant of counting the number of cycles in each of the grid bins [48]. Since there are no points below the $PD$ diagonal, it makes sense to first apply a linear transformation which transforms the multi-set of birth-death ($b, d$) to birth-persistence ($b, d - b$) = ($b, l$) coordinates. Each of the Gaussian functions is centred at a point ($b, l$), with the height of the peak influenced by a given non-negative weight function $\rho : \mathbb{R}^2 \to \mathbb{R}$.

  Typically, $\rho$ reflects some information about the cycles, and it usually depends only on the vertical persistence coordinate $l$ (corresponding to the lifespan of the cycle, $l = d - b$); we choose $\rho(b, l) = l^2$. In our experiments, we consider a grid of size $10 \times 10$, and set the Gaussian function variance to 5% of the maximum death value in $PD$s for the given filtration function and homological dimension. An important advantage of $PI$s is their flexibility, since it is possible to tweak their definition with different grid resolution, weight function, but also different probability density function (and their associated parameters). However, this requirement to make a choice about the $PI$ parameters is also its weakness, since the choice is noncanonical [35].

A more detailed, step-by-step procedure to construct $PL$s and $PI$s from $PD$s can be found in the literature, see, for example, [19] Figs 6 and 7. Figs 5 and 6 show 0- and 1-dimensional $PD$s, $PL$s and $PI$s for an example MNIST image, across selected filtrations.

In order to evaluate the noise robustness of PH, we are interested in computing the distance between PH information of two images. These two images will, for example, be the non-noisy and noisy version of an image. Different metrics can be considered on the space of any persistence signatures. The most common distance between $PD$s is the Wasserstein distance:

$$W_p(PD_1, PD_2) = \inf_\tau \left( \sum_i \|(b_i, d_i) - \tau(b_i, d_i)\|_\infty^p \right)^{\frac{1}{p}}, \tag{1}$$

where the infimum is taken across all bijections $\tau: PD_1 \to PD_2$, and the sum across all

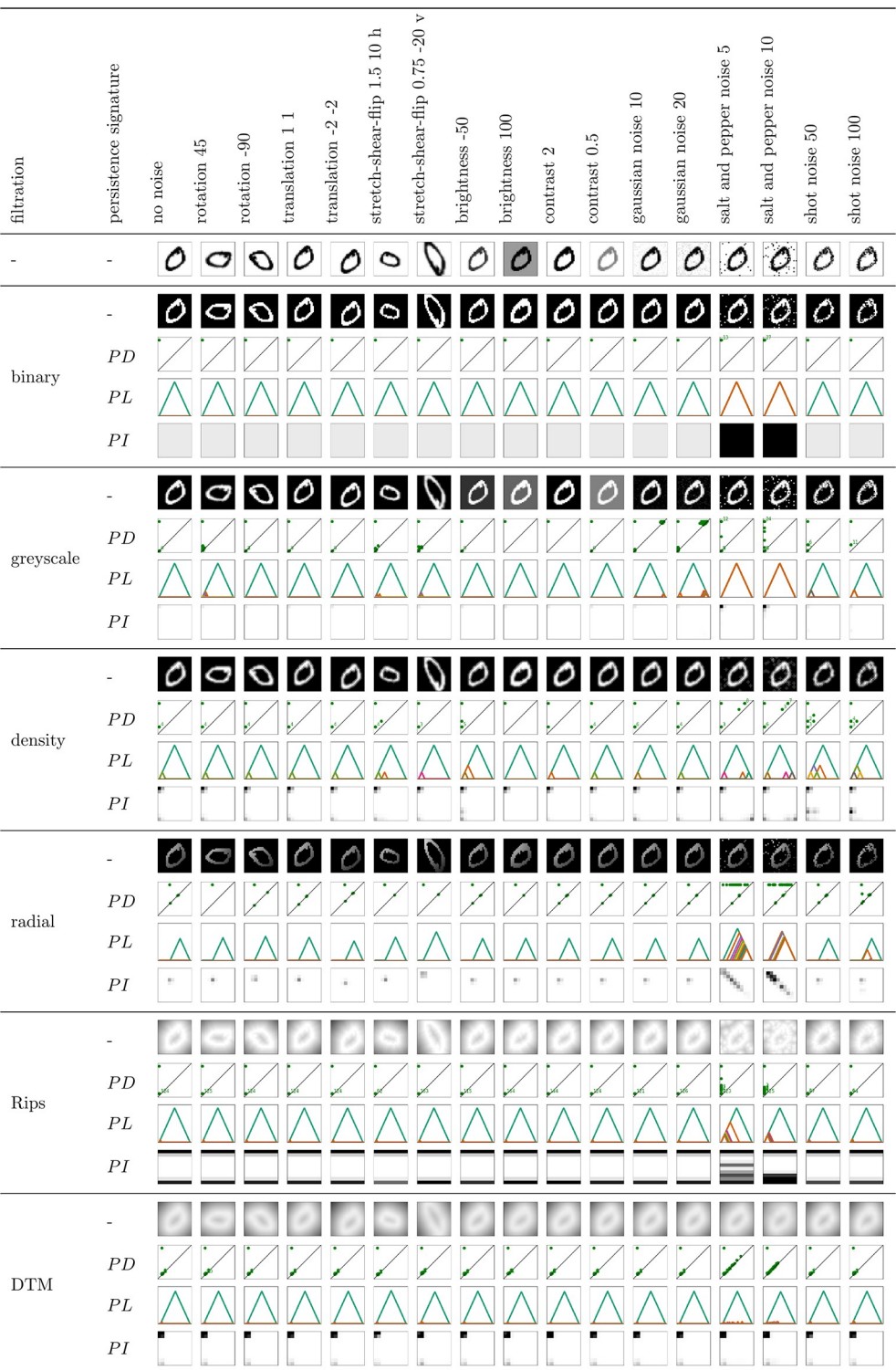

**Fig 5. Noise robustness of 0-dimensional persistent homology on an example image.** Illustration of the effect of various image transformations when the image is represented with its filtration function values (1st row of each filtration), or 0-dimensional persistence diagram (2nd row), persistence landscape (3rd row), or persistence image (4th row).

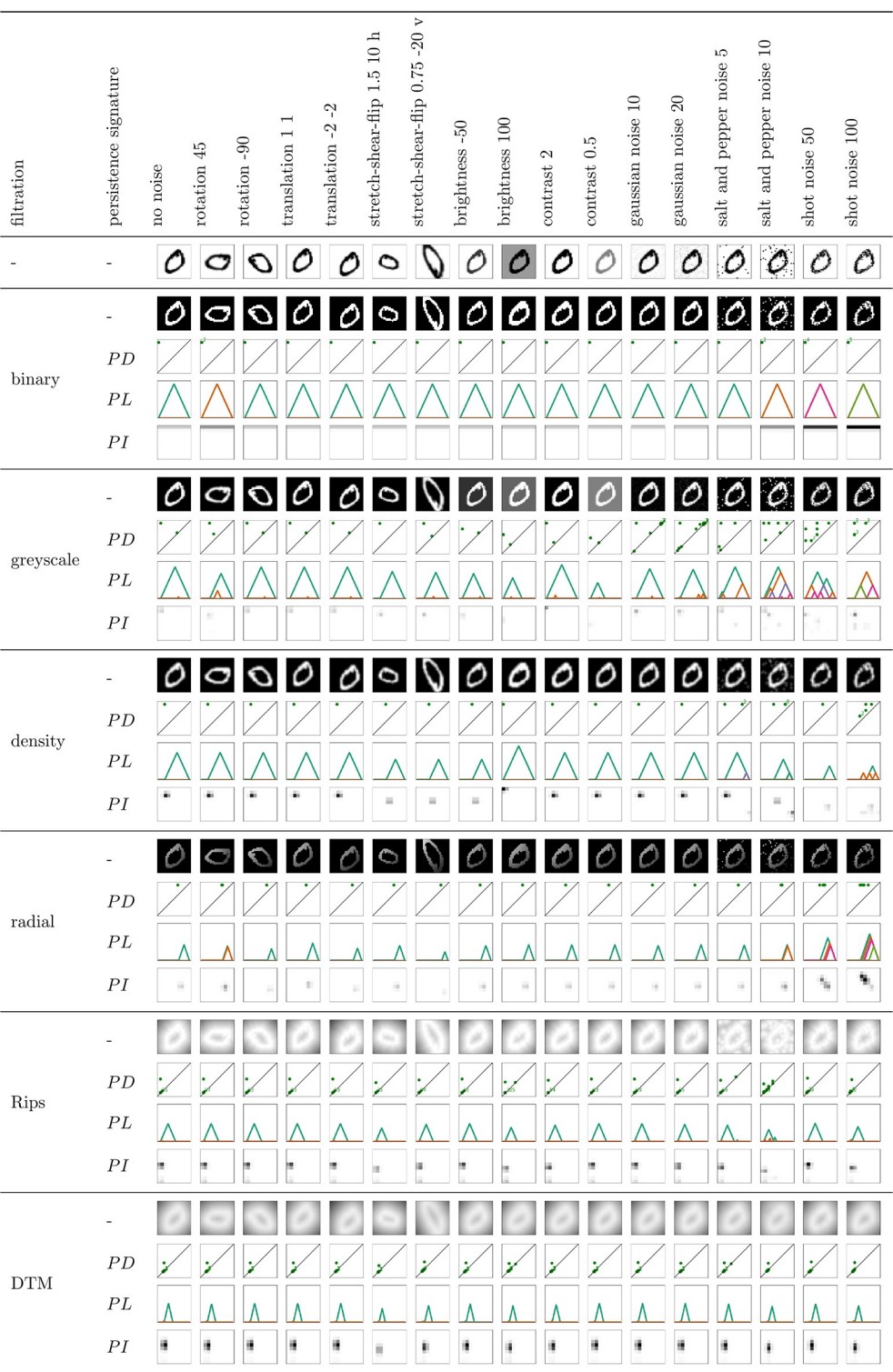

**Fig 6. Noise robustness of 1-dimensional persistent homology on an example image.** Illustration of the effect of various image transformations when the image is represented with its filtration function values (1st row of each filtration), or 1-dimensional persistence diagram (2nd row), persistence landscape (3rd row), or persistence image (4th row).

persistence intervals $(b_i, d_i) \in PD_1$ [42]. There exists a bijection between any two $PD$s, since it is possible to add as many points on the $PD$ diagonal as necessary [45]. This notion of distance is popular in computer vision [49], and it is the common metric for optimal transportation problem [50] (with a bijection $\tau$ from $PD_1$ to $PD_2$ corresponding to a transport plan). In the computational experiments, we will consider the Wasserstein $W_2$ metric between $PD$s, and the $l_2 = \|\cdot\|_2$ metric for the vector persistence signatures $PL$s and $PI$s. The parameter $p$ in both $W_p$ and $l_p$ determines the importance of long compared to short distances.

Furthermore, for a chosen $p$, the choice of persistence signature also influences the importance of cycle lifespans. Indeed, it is easy to see that the Wasserstein $W_p^p$ and $l_p^p$ distances between $PD$s and $PL$s or $PI$s corresponding to $PH_1 = \{(b, d)\}$ and $PH_2 = \emptyset$ reflect $(d - b)^p$, $(d - b)^{p+1}$ and $\rho^p(b, d - b)$. Since we consider $\rho(b, d - b) = (d - b)^2$ as the weight function for persistence images, this means that the cycles that persist for a short time matter the least for $PI$s, and the most for $PD$s (Table 1).

The choice of persistence signature, and the corresponding metric, therefore has an important influence on the noise robustness and discriminative power of PH, although, surprisingly, little research has been carried out in this area before [51]. Recently, [51] evaluated the overlap between the $l_p$ distances between persistence landscapes and persistence images, and the Wasserstein $W_p$ distances between persistence diagrams, on three different datasets (including MNIST images). The results clearly show that the distances between vectorized persistence summaries greatly differ from the distances between $PD$s. Another recent and detailed investigation of the distance correlation between different persistence signatures can be found in [52]: the authors conclude that the considered signatures are "same but different", as they commonly contain the same information, but are shown to yield different results from statistical analyses since they lie in different metric spaces. In addition, the classification accuracy is shown to vary greatly when distances between shapes are given by the distances between their $PD$s, $PL$s or $PI$s in [35, Table 1].

Some other persistence signatures have been introduced in the literature, such as: Betti numbers (across scales) [53, 54]; silhouette [55], which combines all layers of persistence landscape functions into a single, weighted average function, with greater weight assigned to features with longer lifespans [9]; persistence intensity function [56], which is evaluated on a grid to obtain a persistence image [9]; or Euler characteristic, the difference between the number of connected components and the number of holes (across scales) [24]. As already indicated in the Introduction, persistent homology information can also be summarized with a scalar, for instance with: amplitude, distance from empty persistence diagram [36]; entropy, a real number calculated using the lifespans of all features [36], which thus only depends on the persistence but not on the particular birth or death times; or an algebraic function of $b_i$ and $d_i - b_i$,

**Table 1. Persistent homology across signatures.**

| Persistence signature | Limiting behaviour of $\delta^2(PH, \emptyset)$ |
|---|---|
| $PD$ | $O(l^2)$ |
| $PL$ | $O(l^3)$ |
| $PI$, with weight function $\rho(b, l) = l^2$ | $O(l^4)$ |

The choice of persistence signature, and the corresponding metric, determines how sensitive PH is to cycles $(b, d)$ with short persistence, or lifespan, $l = d - b$. The table lists the limiting behaviour, or growth rate, of the function $\delta^2(PH, \emptyset) = \delta^2(\{(b, d)\}, \emptyset) = f(d - b) = f(l)$, where distance $\delta$ represents the Wasserstein $W_2$ distance between persistence diagrams, or $l_2$ distance between persistence landscapes or persistence images. The growth rate reflects the importance of a cycle with lifespan $l$, which influences the noise robustness and discriminative power of PH.

e.g., $\sum b_i^p (d_i - b_i)^q$, so that $p$ and $q$ determine the importance of some of the qualities about cycles (e.g., size of holes) [57]. To avoid the difficult task of choosing among the "zoo of persistence signatures" [52], one can learn the best vector summary of persistence diagram (with e.g., PersLay, a simple neural network layer [58], or ATOL, an unsupervised vectorization method [59]). We do not adopt this approach, as our goal is to illustrate the differences in the noise robustness of PH across signatures. For this purpose, we investigate their behaviour separately, and limit our study to common persistence signatures.

## Stability theorems

Stability theorems are among the most important results in applied and computational topology [42], as they may be viewed as a precise statement about robustness to noise [45]: stable representations of PH are not sensitive to noise in the input.

More precisely, for persistence diagrams calculated with respect to filtration functions $\phi$ and $\psi$, a stability theorem ensures that there exists a constant $c \in \mathbb{R}$ such that:

$$W_p(PD(\phi), PD(\psi)) \leq c \|\phi - \psi\|_p.$$

The stability of *PD*s was first proved for $p = \infty$ (the easiest case, since $W_\infty$ is the least sensitive to details in the diagrams [5]), under some mild conditions on the underlying space $S$ and the filtration functions $\phi$ and $\psi$ [45, 60, 61]. A few years later, the stability was shown to hold for large enough $p$ and under additional assumptions [49], and recently, for any $p$ [42].

A stability theorem for other persistence signatures *PH* states the following:

$$\|PH(\phi) - PH(\psi)\|_p \leq c W_p(PD(\phi), PD(\psi)).$$

Persistence landscapes are shown to be stable for large enough $p$ [47, Theorem 13, Theorem 16], but this fails to be true for $p = 2$ [42, Theorem 7.7]. Stability of persistence images holds for $p = 1$ (under some assumptions on the weight function $\rho$) [35], but not for $p = 2$ [62, Theorem 3], [35, Remark 6].

In the remainder of this section, we discuss the importance of the choice of filtration, signature, and dataset in the interpretation of stability theorems, that is often overlooked in the literature. This discussion then motivates our computational experiments in the next section.

**Stability theorems and the choice of filtration.** The choice of filtration plays a crucial role in the existence and practical value of stability theorems. First of all, in order for the stability theorem to hold for a particular filtration, the filtration function must satisfy the underlying assumptions.

Second, note that the stability theorems ensure that PH is robust under minor perturbations of its input—filtration, and not under minor perturbations in the data space itself. Small changes in the space do not always imply small changes in the filtration function, so that stability theorems provide no guarantee of robustness in such a scenario. For instance, if $Z'$ is obtained by changing the image $Z$ only slightly, $\|\delta_{X(Z,z_0)} - \delta_{X(Z',z_0')}\|_p$ can be large (and it corresponds to the Gromov-Hausdorff distance between point clouds $X(Z, z_0)$ and $X(Z', z_0')$, for $p = \infty$ [63]). Although *PD*s are theoretically stable with respect to the Rips filtration (with the distance function $\delta_{X(Z,z_0)} : \mathbb{R}^n \to \mathbb{R}$ as its filtration function), the upper bound for $W_p(PD(\delta_{X(Z,z_0)}), PD(\delta_{X(Z',z_0')}))$ is so large that it makes little sense in practice: these *PD*s are sensitive to outliers.

Finally, stability theorems are worst-case results, as they do not necessarily ensure tightness of the upper bound provided for the distance between PH information. This is true even if small perturbations in the data result only in small perturbations of the filtration. Let us

consider an image $Z$, and another image $Z'$ obtained with some transformation $\pi: Z \to Z'$. If we apply the stability theorem to the space $Z$ and filtration functions $\phi_{\mathrm{grsc}} : K(Z) \to \mathbb{R}$ and $\psi_{\mathrm{grsc}} = \phi_{\mathrm{grsc}} \circ \pi : K(Z) \to \mathbb{R}$, the right-hand side in the stability theorem $\|\phi_{\mathrm{grsc}} - \psi_{\mathrm{grsc}}\|_p$ (and this change in the greyscale values precisely corresponds to the change in the image) is an upper bound for the change in $PD$s.

If $Z$ is the MNIST image of digit 6, and $Z'$ the same image but with one pixel changed from black to white (Fig 4), then $\|\phi_{\mathrm{grsc}} - \psi_{\mathrm{grsc}}\|_p = 255$ is sufficiently large to allow to change $PD(\phi_{\mathrm{grsc}})$ with one hole to $PD(\psi_{\mathrm{grsc}})$ with no holes. However, if $Z$ is the MNIST image of a digit 0, and $Z'$ the same image but with one pixel changed from white to black (Fig 4), we again have $\|\phi_{\mathrm{grsc}} - \psi_{\mathrm{grsc}}\|_p = 255$ but $PD$ remains unchanged. As another example, we can consider $Z'$ to be the translated image $Z$, when $\|\phi - \psi\|_p$ is large for both the greyscale or radial filtration function. However, $W_p(PD(\phi), PD(\psi))$ is zero when $\phi$ is greyscale (as $PD$s then only register the number and brightness of cycles), but it is large for the radial filtration function (which also captures the position of cycles).

**Stability theorems and the choice of persistence signature.** It is clear from the introduction of this section that stability theorems only hold for some signatures, and some metrics. We already mentioned that $PL$s and $PI$s are shown not to be stable with respect to the $l_2$ metric in [42], although this is the standard choice in applications, when it is commonly assumed that these are stable representations. This is one of the reasons why [42] recently emphasized that "the stability theorems are one of the most misunderstood and miscited results within the field of topological data analysis". It is, however, interesting to see if the stability holds in practice, and to which degree.

**Stability theorems and the choice of dataset.** If the stability theorem holds for a chosen filtration and persistence signature, it does not imply the noise robustness of PH features in a classification task—this depends on the application domain, i.e., the choice of dataset.

Let us go back to the example of $Z$ being the MNIST image of digit 6, and $Z'$ being the same image but with one pixel changed from black to white (Fig 4). As already indicated, the upper bound for the greyscale filtration is large enough to allow to change $PD(\phi_{\mathrm{grsc}})$ with one hole to $PD(\psi_{\mathrm{grsc}})$ with no holes. This is problematic for the classification of the MNIST dataset using $PD$s, since any image contains none, one or two holes, but it would pose less of an issue if there is a greater variety in the number of holes across data classes.

## Results and discussion

We start this section by describing the dataset of greyscale images, and the different types of noise considered in our experiments. In the next subsection, we investigate how sensitive the persistent homology information is to these types of noise, by evaluating the distance between PH for noisy and non-noisy images. This information, however, only paints a part of the picture, since in practical use cases, the PH information must also vary sufficiently among data points in order to form discriminative features in e.g., classification tasks. In the final subsection, we thus investigate the noise robustness of persistent homology together with its discriminative power, by evaluating the drop in classification accuracy when the test data consists of noisy, compared to non-noisy images.

### (Noisy) datasets

We consider the MNIST dataset [64], as it is a well-defined benchmark of greyscale images, and the shape of each of the digits is well understood. To reduce the computation time, we restrict the study to the first 1000 images in the dataset. We investigate three types of affine transformations, changes in image brightness and contrast, and three types of pure noisy

**Table 2. Image noise.**

| Transformation | Definition of transformation |
|---|---|
| rotation | Rotation by 45 degrees clockwise (rotation 45), or 90 degrees counter-clockwise (rotation -90). |
| translation | Translation by 1 pixel right and down (translation 1 1), or 2 pixels left and up (translation -2 -2). |
| stretch, shear and reflect | Stretch, shear and flip respectively by a factor of 1.5 (i.e., an image is scaled down by factor of 1.5 in the $x$ direction, whereas it remains unchanged in the $y$ direction, so that the image is stretched), 10 degrees and horizontal (stretch-shear-flip 1.5 10 h), or by a factor 0.75, -20 degrees and vertical (stretch-shear-flip 0.75 -20 v). |
| brightness | -50 or 100 is added to the greyscale value of each pixel (brightness -50 and brightness 100, respectively). |
| contrast | Greyscale value of each pixel is multiplied with 2 or 0.5 (contrast 2 and contrast 0.5, respectively). |
| gaussian noise | Random noise drawn from normal distribution $\mathcal{N}(0, 10)$ or $\mathcal{N}(0, 20)$ is added to the greyscale value of each pixel (gaussian noise 10 and gaussian noise 20, respectively). |
| salt and pepper noise | 5% or 10% of random pixels in an image are changed, with equal probability, to either white (i.e., salt) or black (i.e., pepper) (salt and pepper noise 5 and salt and pepper noise 10 respectively). |
| shot noise | Greyscale value of each pixel is replaced with a random number drawn from the Poisson distribution, with the distribution mean corresponding to the original greyscale pixel value, scaled down with a factor 50 (shot noise 50) or 100 (shot noise 100), as Poisson distribution is spread out more for lower means. Since the Poisson distribution with mean zero is equal to zero, the shot noise only changes non-white pixels. |

transformations, each at two different levels, and in different directions, if applicable (Table 2). The greyscale pixel values are clipped to the interval [0, 255].

For every (non-noisy and noisy) dataset, i.e., for each image in each of the datasets, we calculate the values of filtration functions on each pixel, and the 0- and 1-dimensional persistent homology information with respect to all considered filtrations and persistence signatures (with the specified values of the parameters) using the python GUDHI library [65]. For 0-dimensional homology, we truncate the death value of infinite intervals to the maximum finite death value for the given filtration function, across all transformations.

## Noise robustness

The goal of this section is to understand in what way, and to which degree, is the persistent homology information sensitive to noise, across different filtrations and persistence signatures. In order to address this question, we start by visualizing the different filtration functions and persistent homology information for an *example* MNIST image, under different data transformations (Figs 5 and 6). We can conclude the following.

- affine transformations (rotation, translation, stretch-shear-reflect): PH on binary and greyscale filtration remains unchanged under any affine transformations, since it only registers the number and brightness of cycles (it is a topological invariant). Note, however, that the affine transformations sometimes slightly disturb the greyscale values in the computational experiments, so that, e.g., some cycles can appear or disappear in an image (see, for instance, the additional one-pixel hole for the binary filtration under rotation 45 in Fig 6). Under stretch-shear-reflect, the density along and within a hole changes, which results in a change of birth and death values for 1-dimensional cycles with respect to the density filtration. Radial filtration function captures the position of the cycles, so that the birth and death values of cycles can also change significantly under any affine transformation. PH on

Rips and DTM filtration is robust under rotation and translation. However, PH with Rips and DTM filtration captures the size of cycles, and is thus sensitive to affine transformations that rescale the image. For instance, under stretch-shear-reflect that enlarges a digit, the number of point cloud points increases, resulting in many additional short persisting 0-dimensional cycles for these filtrations. The death value of 1-dimensional cycles for Rips and DTM filtration also changes under stretch-shear-flip, as the PH in this case reflects the size of the hole.

- **brightness**: PH on binary and radial filtration does not see important changes if the brightness of an image is adjusted. However, a change in image brightness does result in changes of the birth or death values in 1-dimensional PH on greyscale or density filtration, and additional cycles can be captured with density filtration. A change of thickness of a digit also results in additional 0-dimensional cycles for Rips and DTM filtration, that are of short persistence, but there are many. For these filtrations, there is also a minor change in the death value for 1-dimensional cycles, as it captures the size of the hole that can change under a change in brightness.

- **contrast**: PH with respect to most of the considered filtrations is invariant under changes in the contrast of an image. The only exception is 1-dimensional PH with greyscale filtration, where the birth or death value of cycles can change.

- **salt and pepper noise**: Gaussian, salt and pepper, and shot noise change the greyscale value of some random pixels. For each black pixel on a white background in the salt and pepper noise, a new one-pixel connected component (a long persisting 0-dimensional cycle) appears for PH on binary, greyscale, and radial filtration. If a pixel in a neighbourhood of black pixels is changed to white, an additional long persisting 1-dimensional cycle (one-pixel hole) can appear for PH on these filtrations. Also, an existing hole in the non-noisy image may become disconnected in the noisy image, and thus not registered. The additional 0-dimensional cycles are all born at birth value 0 for PH on Rips filtration, but they die earlier (as soon as they are connected to another point cloud point), so that Rips is more robust under this transformation, but still severely impacted by the outliers. One-pixel or disconnected holes are not an issue for PH on Rips filtration, but the death value of 1-dimensional cycles can decrease due to the additional pixels within a hole (see also the image of digit 0, and the same image with a single outlier in Fig 4). PH on DTM filtration is significantly more robust to salt and pepper noise, as the outliers are "washed out".

- **gaussian noise**: The gaussian noise produces similar type of perturbations as the salt and pepper noise, but the change in the greyscale value is much less prominent, so that no additional cycles are typically seen with the binary or radial filtration (which take the binary image as input), and have a very low persistence for the greyscale filtration.

- **shot noise**: Shot noise only changes the non-white pixels (to lighter or darker), so that a digit might become disconnected into a few components, a hole might become disconnected, and many one-pixel holes may appear. The additional 0-dimensional cycles have a long lifespan for binary and radial filtration, but they are short for PH on greyscale filtration (or more precisely, they are directly related to the strength of the change of the greyscale pixel values) and density filtration. 1-dimensional PH with these filtrations exhibits similar behaviour. As already mentioned, PH with Rips and DTM filtration is more robust under this type of noise, since disconnected components or holes can still be captured, as the Rips and DTM filtration connect non-neighbouring pixels with a sufficiently large edge (resolution $r$ in the filtration).

*PD*s, *PL*s and *PI*s reflect the same information about the cycles, and Figs 5 and 6 show that they change accordingly. However, without considering the metric on these spaces of persistence signatures, we cannot derive any insights regarding the difference in the noise robustness from these figures.

Furthermore, the major part of the discussion above is based only on a single example data point. We therefore calculate the ($l_2$ or $W_2$) distance between each image in the dataset, and its noisy variant, when images are represented with their filtration functions or persistent homology information (Table 3). The results on the complete dataset align well with the findings discussed above for an example image. In addition, Table 3 clearly shows that, for any given filtration and homological dimension, there is a relative difference between *PD*s, *PL*s and *PI*s in robustness under various transformations. For instance, 0-dimensional PH on Rips filtration is more sensitive to salt and pepper than shot noise for any persistence signature, but this difference is much more pronounced for *PL*s, and in particular for *PI*s, compared to *PD*s.

Finally, Table 3 implies that stability theorems do not necessarily provide useful information about the stability in practice. For example, under rotation and gaussian noise, the average value of $\|\phi_{\mathrm{grsc}} - \psi_{\mathrm{grsc}}\|_2$ is respectively equal to 2707.85 and 412.55. However, we see that the distance between PH on noisy and non-noisy images is close to zero for rotation, but it is much larger under gaussian noise.

## Noise robustness and discriminative power

In the previous section, we assess the distances between images and their noisy version. In practical applications, however, these distances ought to be compared to the distances between the images in (other classes of) the dataset, which reflect the discriminative power in a classification task. Therefore, in this section, we discuss the noise robustness together with the discriminative power of persistent homology, across different filtrations and persistence signatures, for non-noisy and noisy datasets.

In order to do so, we investigate how much the performance of a classifier (more specifically, a Support Vector Machine (SVM)) drops when noise is added to the dataset. Since *PD*s are multi-sets, we use an SVM with a gaussian kernel:

$$\kappa(Z, Z') = e^{-\frac{\delta^2(Z,Z')}{2\sigma^2}}.$$

For two images $Z$ and $Z'$, $\delta(Z, Z')$ corresponds to the Wasserstein $W_2$ distance between their *PD*s, or the $l_2$ distance between their filtration function values, *PL*s or *PI*s. Note that the space of *PD*s with Wasserstein metric is not of negative type [52, Theorem 3.2], so that this kernel is not an inner product [66].

For each representation of the images, the SVM regularization parameter (typically noted as *C*, which trades off correct classification of training examples against maximization of the decision function's margin) and the kernel parameter $\sigma^2$ are first tuned using 5-fold cross validation on the training set of 70% non-noisy images. We consider $C \in \{10^{-1}, 10^0, 10^1, 10^2\}$, and $\gamma = \frac{1}{2\sigma^2} \in \{10^{-7}, 10^{-6}, 10^{-5}, 10^{-4}, 10^{-3}, 10^{-2}\}$. As we are focused on noise robustness, we calculate the relative decrease in accuracy for noisy compared to non-noisy test data (the remaining 30% of images in the dataset). The results are summarized in Fig 7.

We observe that there is at least 35% drop in SVM accuracy, when images are represented with PH, in the following scenarios:

- affine transformations: PH on radial filtration under any affine transformation, and PH on Rips and DTM for stretch-shear-flip.

- brightness: PH on greyscale, Rips and DTM filtration.

**Table 3. Noise robustness of persistent homology on 1000 MNIST greyscale images.**

| filtration | homological dimension | persistence signature | rotation 45 | rotation -90 | translation 1 1 | translation -2 -2 | stretch-shear-flip 1.5 10 h | stretch-shear-flip 0.75 -20 v | brightness -50 | brightness 100 | contrast 2 | contrast 0.5 | gaussian noise 10 | gaussian noise 20 | salt and pepper noise 5 | salt and pepper noise 10 | shot noise 50 | shot noise 100 |
|---|---|---|---|---|---|---|---|---|---|---|---|---|---|---|---|---|---|---|
| binary | - | - | 11.4 | 11.95 | 9.12 | 11.9 | 10.59 | 12.28 | 2.73 | 5.63 | 4.31 | 0.00 | 1.45 | 2.11 | 4.42 | 6.24 | 5.0 | 6.34 |
|  | 0 | PD | 0.01 | 0.00 | 0.00 | 0.00 | 0.02 | 0.01 | 0.01 | 0.02 | 0.01 | 0.00 | 0.00 | 0.00 | 1.80 | 2.43 | 0.17 | 0.64 |
|  |  | PL | 0.06 | 0.00 | 0.00 | 0.01 | 0.16 | 0.10 | 0.09 | 0.18 | 0.12 | 0.00 | 0.04 | 0.03 | 12.03 | 12.17 | 1.42 | 5.16 |
|  |  | PI | 10.82 | 0.00 | 0.00 | 1.91 | 27.37 | 15.92 | 17.19 | 30.56 | 21.01 | 0.00 | 5.73 | 5.73 | 8430.76 | 15135.00 | 282.02 | 1395.47 |
|  | 1 | PD | 0.04 | 0.00 | 0.00 | 0.00 | 0.05 | 0.05 | 0.03 | 0.11 | 0.08 | 0.00 | 0.01 | 0.02 | 0.32 | 0.52 | 0.67 | 0.66 |
|  |  | PL | 0.33 | 0.00 | 0.00 | 0.00 | 0.37 | 0.40 | 0.27 | 0.91 | 0.64 | 0.00 | 0.10 | 0.17 | 2.61 | 4.23 | 5.45 | 5.33 |
|  |  | PI | 18.59 | 0.00 | 0.00 | 0.00 | 20.20 | 20.20 | 14.14 | 51.72 | 35.36 | 0.00 | 5.05 | 8.89 | 170.32 | 333.36 | 503.68 | 494.59 |
| grsc | - | - | 2454.9 | 2707.85 | 1906.79 | 2679.12 | 2335.78 | 2656.99 | 1268.72 | 2646.74 | 653.29 | 3295.03 | 207.45 | 412.55 | 1104.09 | 1560.62 | 814.52 | 1092.34 |
|  | 0 | PD | 22.23 | 0.04 | 0.00 | 0.34 | 26.36 | 24.21 | 1.64 | 13.44 | 13.64 | 12.78 | 46.77 | 92.04 | 454.15 | 611.47 | 90.59 | 125.29 |
|  |  | PL | 65.97 | 0.18 | 0.00 | 1.89 | 94.07 | 73.28 | 8.72 | 50.54 | 53.11 | 44.99 | 88.45 | 236.47 | 3059.20 | 3102.04 | 449.45 | 710.91 |
|  |  | PI | 2.46 | 0.00 | 0.00 | 0.10 | 3.44 | 2.78 | 0.66 | 1.88 | 2.33 | 1.69 | 5.67 | 19.13 | 795.05 | 1427.76 | 23.79 | 44.92 |
|  | 1 | PD | 19.29 | 0.00 | 0.00 | 0.00 | 22.02 | 21.96 | 22.24 | 46.86 | 26.27 | 52.87 | 17.31 | 34.23 | 77.16 | 118.86 | 97.97 | 132.88 |
|  |  | PL | 125.50 | 0.00 | 0.00 | 0.00 | 147.35 | 133.64 | 165.39 | 314.50 | 162.01 | 331.13 | 58.95 | 121.08 | 516.37 | 810.41 | 532.87 | 803.11 |
|  |  | PI | 19.24 | 0.00 | 0.00 | 0.00 | 18.80 | 18.97 | 20.95 | 20.49 | 19.09 | 19.23 | 13.05 | 19.03 | 29.18 | 49.70 | 32.54 | 57.68 |
| density | - | - | 43.71 | 46.95 | 28.47 | 44.68 | 39.94 | 47.96 | 6.65 | 15.38 | 11.08 | 0.00 | 3.29 | 4.8 | 10.1 | 14.79 | 12.12 | 17.68 |
|  | 0 | PD | 0.60 | 4.17 | 1.67 | 3.17 | 3.74 | 6.46 | 0.61 | 0.78 | 0.70 | 0.00 | 0.29 | 0.41 | 1.75 | 2.35 | 1.52 | 2.10 |
|  |  | PL | 2.40 | 27.58 | 11.64 | 21.85 | 23.97 | 44.58 | 2.51 | 3.13 | 2.78 | 0.00 | 1.15 | 1.62 | 7.35 | 10.66 | 7.15 | 11.04 |
|  |  | PI | 6.10 | 23.74 | 16.06 | 23.37 | 19.76 | 33.39 | 6.96 | 8.60 | 7.30 | 0.00 | 2.42 | 3.58 | 22.52 | 33.69 | 21.81 | 35.23 |
|  | 1 | PD | 0.32 | 1.37 | 0.59 | 1.18 | 1.51 | 2.56 | 0.25 | 0.60 | 0.43 | 0.00 | 0.09 | 0.17 | 0.66 | 1.21 | 0.69 | 1.06 |
|  |  | PL | 1.78 | 7.33 | 3.60 | 6.51 | 8.36 | 13.99 | 1.44 | 3.75 | 2.58 | 0.00 | 0.51 | 0.94 | 2.84 | 5.04 | 3.73 | 5.39 |
|  |  | PI | 7.04 | 4.12 | 3.23 | 4.25 | 10.41 | 7.81 | 5.64 | 16.54 | 11.27 | 0.00 | 1.95 | 3.64 | 8.32 | 17.30 | 12.49 | 14.21 |
| radial | - | - | 205.27 | 216.86 | 172.73 | 202.44 | 196.24 | 233.76 | 49.75 | 102.54 | 78.48 | 0.00 | 25.93 | 38.34 | 84.06 | 119.18 | 90.77 | 115.35 |
|  | 0 | PD | 3.30 | 4.17 | 1.67 | 3.17 | 3.74 | 6.46 | 0.48 | 1.19 | 0.86 | 0.00 | 0.20 | 0.29 | 34.17 | 46.66 | 3.84 | 12.07 |
|  |  | PL | 19.65 | 27.58 | 11.64 | 21.85 | 23.97 | 44.58 | 2.20 | 6.59 | 4.40 | 0.00 | 0.86 | 1.12 | 214.34 | 277.03 | 20.39 | 68.70 |
|  |  | PI | 18.11 | 23.74 | 16.06 | 23.37 | 19.76 | 33.39 | 1.57 | 5.97 | 3.60 | 0.00 | 0.51 | 0.87 | 92.99 | 154.38 | 6.78 | 21.56 |
|  | 1 | PD | 1.44 | 1.37 | 0.59 | 1.18 | 1.51 | 2.56 | 0.57 | 2.00 | 1.36 | 0.00 | 0.22 | 0.35 | 5.36 | 8.83 | 11.45 | 11.37 |
|  |  | PL | 8.07 | 7.33 | 3.60 | 6.51 | 8.36 | 13.99 | 3.00 | 10.82 | 7.31 | 0.00 | 1.19 | 1.88 | 29.14 | 48.57 | 63.27 | 62.90 |
|  |  | PI | 4.40 | 4.12 | 3.23 | 4.25 | 10.41 | 6.26 | 1.01 | 4.01 | 2.59 | 0.00 | 0.39 | 0.66 | 9.44 | 17.20 | 24.42 | 24.86 |
| Rips | - | - | 75.71 | 95.39 | 28.75 | 55.67 | 81.14 | 84.12 | 6.55 | 14.88 | 10.92 | 0.00 | 3.3 | 4.79 | 89.41 | 106.4 | 10.27 | 12.58 |
|  | 0 | PD | 0.82 | 0.01 | 0.01 | 0.09 | 2.88 | 2.85 | 1.39 | 2.85 | 2.18 | 0.00 | 0.50 | 0.63 | 7.72 | 8.48 | 1.95 | 3.29 |
|  |  | PL | 0.21 | 0.00 | 0.00 | 0.07 | 0.37 | 0.25 | 0.24 | 0.46 | 0.31 | 0.00 | 0.11 | 0.09 | 35.45 | 30.59 | 2.11 | 4.77 |
|  |  | PI | 2.96 | 0.01 | 0.02 | 0.37 | 29.10 | 29.33 | 7.04 | 28.41 | 16.83 | 0.00 | 1.40 | 1.85 | 129.83 | 163.86 | 10.19 | 22.72 |
|  | 1 | PD | 0.51 | 0.00 | 0.00 | 0.02 | 1.21 | 1.05 | 0.64 | 1.34 | 1.01 | 0.00 | 0.25 | 0.34 | 1.15 | 1.80 | 1.21 | 1.56 |
|  |  | PL | 0.83 | 0.00 | 0.00 | 0.00 | 2.11 | 1.27 | 0.69 | 1.83 | 1.22 | 0.00 | 0.25 | 0.42 | 3.16 | 5.21 | 1.85 | 2.59 |
|  |  | PI | 0.68 | 0.01 | 0.00 | 0.00 | 2.12 | 1.68 | 0.68 | 2.05 | 1.28 | 0.00 | 0.24 | 0.38 | 1.71 | 2.77 | 1.56 | 2.46 |
| DTM | - | - | 66.74 | 86.45 | 25.79 | 51.24 | 70.8 | 73.56 | 4.35 | 7.6 | 5.99 | 0.00 | 2.14 | 3.08 | 22.51 | 36.64 | 6.77 | 9.22 |
|  | 0 | PD | 1.37 | 0.01 | 0.01 | 0.10 | 3.73 | 4.06 | 1.57 | 2.97 | 2.16 | 0.00 | 0.81 | 1.05 | 4.00 | 5.88 | 2.21 | 3.30 |
|  |  | PL | 1.22 | 0.00 | 0.01 | 0.08 | 5.28 | 4.78 | 1.60 | 4.19 | 2.69 | 0.00 | 0.64 | 0.87 | 6.33 | 9.94 | 2.82 | 5.52 |
|  |  | PI | 1.66 | 0.00 | 0.00 | 0.12 | 14.20 | 12.01 | 3.38 | 12.00 | 7.33 | 0.00 | 0.81 | 1.08 | 6.89 | 13.81 | 4.93 | 10.58 |
|  | 1 | PD | 0.60 | 0.00 | 0.00 | 0.02 | 1.22 | 1.23 | 0.59 | 1.08 | 0.83 | 0.00 | 0.31 | 0.41 | 0.85 | 1.16 | 0.83 | 1.08 |
|  |  | PL | 0.86 | 0.00 | 0.00 | 0.02 | 2.32 | 2.26 | 0.79 | 1.32 | 1.01 | 0.00 | 0.38 | 0.55 | 1.22 | 1.92 | 1.39 | 2.05 |
|  |  | PI | 0.23 | 0.00 | 0.00 | 0.00 | 0.60 | 0.56 | 0.18 | 0.36 | 0.25 | 0.00 | 0.09 | 0.14 | 0.30 | 0.52 | 0.42 | 0.65 |

The table shows the distance $\|\phi - \psi\|_2$ between the filtration function values on the non-noisy and noisy image (1st row of each filtration), the Wasserstein distance $W_2(PD(\phi), PD(\psi))$ between 0- or 1-dimensional persistence diagrams (2nd and 5th row), the distance $\|PL(\phi) - PL(\psi)\|_2$ between persistent landscapes (3rd and 6th row), or the distance $\|PI(\phi) - PI(\psi)\|_2$ between persistent images (4th and 7th row), averaged across 1000 images in the MNIST dataset.

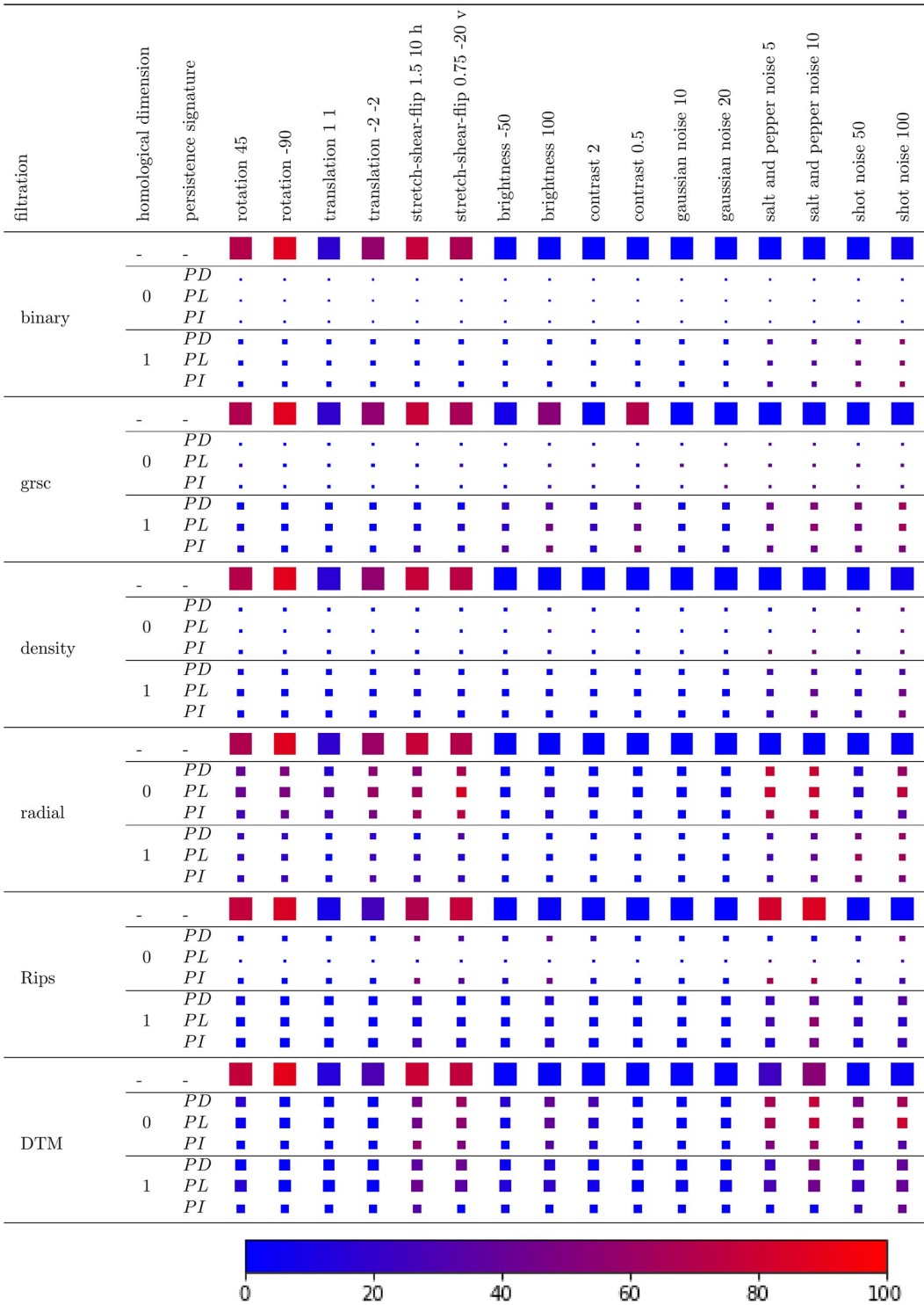

**Fig 7. Noise robustness and discriminative power of persistent homology on 1000 MNIST greyscale images.** The figure shows the drop in SVM classification accuracy when the test dataset is noisy, compared to the non-noisy test set, averaged across 1000 images in the MNIST dataset. Each image is represented either with its filtration function values (1st row of each filtration), or with its 0- or 1-dimensional persistence diagram (2nd and 5th row), persistent landscape (3rd and 6th row) or persistent image (4th and 7th row). The size of the node reflects the absolute accuracy on the non-noisy test data. The colour of the node reflects the accuracy drop, indicated in the colour bar. In particular, the presence of red nodes for PH information (2nd to 7th row) implies that PH is not robust under any type of noise, for any filtration and persistence signature.

- contrast: PH on greyscale filtration.

- gaussian, salt and pepper, shot noise: There is a drop in SVM accuracy for all filtrations under salt and pepper or shot noise. For gaussian noise, the drop in accuracy is negligible for PH on all but the greyscale filtration.

The drop in accuracy also varies for different persistence signatures. For example, 1-dimensional *PL*s on Rips filtration are more sensitive to salt and pepper noise than *PD*s.

We note, however, that if the classification accuracy on the non-noisy data is low, the loss in performance is limited. For instance, 0-dimensional PH with respect to the binary filtration yields an accuracy of only 10% (not better than a random guess), as it only counts the number of components (Fig 4), and every digit 0–9 commonly consists of a single component. This is why there is no drop in accuracy when SVM is tested on images under salt and pepper noise (Fig 7), although this transformation results in an image with many additional connected components (Fig 5) and thus significantly changes PH on binary filtration. In these cases, however, the drop in accuracy gives us no reliable information about noise robustness.

When images are represented with their filtration function values on each pixel (including thus the original representation of an image as a vector of greyscale pixel values), the SVM performance is significantly worse for test data consisting of images with affine transformations. However, the performance is relatively stable under changes in image brightness or contrast, or under noisy transformations (with some exceptions). This is an opposite trend compared to PH, which is often robust under affine, but sensitive under noisy transformations (with a significant difference across filtrations and persistence signatures). Even though PH is often reputed for its robustness to noise [36], if data is expected to contain gaussian, salt and pepper or shot noise, the raw representation of images with their greyscale pixel values is robust to noise (there is no drop in SVM accuracy compared to non-noisy data), while this is often not the case for PH features.

Moreover, the absolute SVM accuracy on non-noisy data, when images are summarized with any persistence signature with respect to any filtration cannot compare with the representation of an image with its filtration function values, which contains more detailed geometric information about the image. Indeed, persistent homology only captures information about cycles in an image, and for most of the filtrations, it can only differentiate between two and three classes among the ten MNIST digits 0–9. The classification accuracy can be significantly improved by concatenating PH across different signatures, homological dimensions and/or filtrations (e.g., a combination of PH on radial and Rips filtration captures both information about the position and size of cycles, and can thus discriminate better across classes). For instance, a set of only 28 features obtained from concatenated persistent homology information is shown in [36] as sufficient to attain better classification accuracy than the set of greyscale pixel values. An alternative approach to simultaneously exploit PH from multiple filtrations is multi-dimensional persistence [67]. Since our goal is to gain insight into the noise robustness of individual PH representations, rather than maximizing the performance of classifiers, such an analysis is out of the scope of this work. However, under some types of transformations, the SVM accuracy is better for some PH features (even without concatenation across filtrations or signatures) compared to the raw representation with greyscale pixel values.

## Conclusions, limitations and future research

Persistent homology, information about connected components, holes, and cycles in higher dimensions, is commonly characterized in the literature as a *topological* summary *robust to noise*. The main motivation behind this paper is to illustrate how misleading this description

can be, particularly for practical applications. We show that the validity of such a characterization, in theory, depends strongly on the choice of filtration and persistence signature (input and output of PH), and in practice, also on the particular application domain.

First of all, we emphasize that the type of information PH captures about cycles, is determined by the choice of filtration. For some filtrations, this information is only of *topological* nature, but for others, some *geometric* information can be captured as well. Topological invariants are robust under affine transformations, but the same does not necessarily hold for geometric invariants, so that the choice of filtration directly influences the noise robustness of PH.

Moreover, we underline how stability theorems, which provide a theoretical guarantee of the noise robustness of PH, depend on the choice of filtration and persistence signature, as well as the distance metric between them. Firstly, stability theorems make some assumptions about the filtration function, e.g., the function must be tame (the corresponding persistence diagram has finitely many off-diagonal points [60]), monotonic, continuous, Lipschitz or piecewise constant. Secondly, the robustness of PH is only guaranteed under small changes of the input—the filtration, rather than small changes in the space itself. For instance, if one background white pixel in an image is changed to black, the distance between the filtration functions for the common Vietoris-Rips filtration between these two images is large, and indeed, PH with respect to the Rips filtration is sensitive to such outliers. Furthermore, the statement of stability theorems is restricted to the particular choice of persistence signature and metric. This is often overlooked in the literature: e.g., it is common to employ persistence landscapes or persistence images and the euclidean metric, whereas the stability theorems do not hold in such scenarios.

Finally, even if a stability theorem holds for the particular choice of filtration and persistence signature, it does not imply that PH yields noise-robust features in a classification task—this is domain and application-specific. For instance, changing a single pixel in an image from black to white can result in an additional one-pixel hole, which can be persistent for some filtrations. This change in PH is substantial if the number of holes does not vary greatly across classes of data, when the presence of such noise can be expected to deteriorate the classification accuracy. Reversely, even if there is no theoretical guarantee of the stability of PH for the given filtration and signature, it is interesting to evaluate the noise robustness in practice. To gain a better understanding of the noise robustness of PH, we carry out some computational experiments on the well-known MNIST dataset of greyscale images, under some common types of noise to be expected on such data. We conclude that there is a considerable drop in accuracy of SVM trained on PH information of non-noisy and tested on noisy data, for at least 0- or 1-dimensional PH, for at least one of the considered signatures:

- rotation and translation: radial

- stretch-shear-flip: radial, Rips, DTM

- brightness: greyscale, Rips, DTM

- contrast: greyscale

- gaussian noise: greyscale

- salt and pepper, and shot noise: binary, greyscale, density, radial, Rips, DTM.

There is often also an important difference in the drop in accuracy across *PD*s, *PL*s and *PI*s. Taking all the above into consideration, it is clear that one needs to be more careful when referring to persistent homology as a noise-robust topological invariant: this is only true for some filtrations and signatures, and even in such cases, the stability of PH does not necessarily imply that the presence of noise will not weaken the discriminative power of PH features.

The main findings of this paper provide some guidelines on the choice of suitable filtration(s) and persistence signature(s), and the corresponding metric, for the given dataset and expected types of noise. Some important questions that should be addressed when using persistent homology are the following:

- choice of filtration: What information about cycles (number, brightness, position, size) is different across classes of data, but does not change much for the expected type of noise? Does the filtration function satisfy the assumptions in the stability theorem? Do small changes in the data result in small changes in the filtration function?

- choice of persistence signature: Is the signature stable? Are the cycles with the longest persistence or lifespan the most important (i.e., should cycles with short lifespan be considered as noise)? If this is not the case, it is a good idea to use a flexible signature which allows cycles of e.g., medium persistence to be the most crucial (such as *PI*s with an appropriate weight function, what is not immediately possible with *PD*s or *PL*s). How critical are the important compared to unimportant cycles? Which statistical or machine learning methods do we want to apply to PH? If PH does not need to be summarized as a function or vector, it might be sufficient to use *PD*s. How important is computational efficiency? If the computation time is limited, it might be useful to avoid *PD*s and the expensive calculation of Wasserstein distances.

- choice of metrics: Is the signature stable? How critical are the important compared to unimportant cycles? The greater the $p$, the bigger is the difference across cycles, for both Wasserstein $W_p$ or $l_p$ metric, i.e., PH is less sensitive to unimportant cycles.

In summary, the choice of filtration defines the persistence of different types of cycles (e.g., for PH with Rips filtration, small cycles have short persistence), the choice of signature defines which cycles are least important or noisy (e.g., these are typically the cycles with short persistence), and together with the choice of metric determines the level of sensitivity to noisy cycles.

Our findings are limited to the particular setting in our computational experiments: the choice of filtrations and persistence signatures, and their parameters, the choice of metric, dataset, noise, and classifier. For future research, it would be interesting to revisit similar research questions, but in a different context, e.g., for a different dataset. The MNIST images of digits 0–9 all typically have one connected component, and none, one or two holes. Both noise robustness and classification accuracy are expected to be better for datasets where the number (but also other properties such as size) of cycles differ greatly across classes, such as images with complex structure which come from materials science, astronomy, neuroscience, plant morphology (e.g., images of cosmic web, protein networks, brain arteries, plant roots).

## Supporting information

**S1 Appendix.**
(PDF)

## Author Contributions

**Conceptualization:** Renata Turkeš.

**Data curation:** Renata Turkeš, Jannes Nys.

**Formal analysis:** Renata Turkeš, Jannes Nys, Tim Verdonck.

**Funding acquisition:** Steven Latré.

**Investigation:** Renata Turkeš.

**Methodology:** Renata Turkeš, Jannes Nys.

**Project administration:** Steven Latré.

**Resources:** Steven Latré.

**Software:** Renata Turkeš.

**Supervision:** Jannes Nys.

**Validation:** Renata Turkeš.

**Visualization:** Renata Turkeš, Jannes Nys.

**Writing – original draft:** Renata Turkeš.

**Writing – review & editing:** Renata Turkeš, Jannes Nys, Tim Verdonck, Steven Latré.

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
