## [Decision Letter · Decision Letter 0]

30 Jun 2021

PONE-D-21-14381

Noise robustness of persistent homology on greyscale images, across filtrations and signatures

PLOS ONE

Dear Dr. Turkes,

Thank you for submitting your manuscript to PLOS ONE. After careful consideration, we feel that it has merit but does not fully meet PLOS ONE’s publication criteria as it currently stands. Therefore, we invite you to submit a revised version of the manuscript that addresses the points raised during the review process.

Both reviewers found the paper and the results quite interesting. The comments mostly focused on improving the clarity of the text and some notational issues. These relatively minor issues should be addressed before publication. 

We look forward to receiving your revised manuscript.

Kind regards,

Giovanni Petri, Ph.D.

Academic Editor

PLOS ONE

Additional Editor Comments (if provided):

Reviewers' comments:

Reviewer's Responses to Questions

**Comments to the Author**

1. Is the manuscript technically sound, and do the data support the conclusions?

Reviewer #1: Yes

Reviewer #2: Yes

2. Has the statistical analysis been performed appropriately and rigorously? 

Reviewer #1: Yes

Reviewer #2: Yes

3. Have the authors made all data underlying the findings in their manuscript fully available?

Reviewer #1: Yes

Reviewer #2: Yes

4. Is the manuscript presented in an intelligible fashion and written in standard English?

Reviewer #1: Yes

Reviewer #2: Yes

5. Review Comments to the Author

Reviewer #1: The paper is interesting and deals with relevant issues in TDA.

Our suggestion and observations are included in the attached pdf. There is a list of minor improvements after which I thin the paper will be suitable for publication on PLOS.

Reviewer #2: The paper presents how noise in data affects its persistent homology. It clearly explains the relevance of the problem and how the results compare with existing literature.

The authors discuss how different choices of filtrations and noise on the MNIST handwritten digits dataset perturb the persistent signatures (persistence diagram, persistent images and persistent landscapes). This presents a useful practical example of how the theoretical stability theorem might or not work in practice. The authors provide exhaustive computational results of why in practice it is not easy to control the robustness to noise of the persistence signatures. Moreover, they provide additional theoretical explanations of how the stability theorem could lead to misleading claims if wrongly interpreted. Finally, the different persistent signatures are used as an input of a classifier to quantify the loss in accuracy in the noisy data.

For all these reasons, I would accept this paper after the following minor revisions.

1. In the definition of abstract simplicial and cubical complex. The definitions are given in a very intuitive way but it would be useful to mention that they are closed under taking subsets (i.e if we have a triangle all the edges and vertices are present).

2. It might be worth mentioning that representing each pixel with a cube is not the unique representation of an image as a cubical complex. One might also consider the dual of the complex.

3.It might be useful adding a toy example describing a cubical filtration and what are cavities in a cubical complex.

4. It might be useful adding a toy example of different filtrations. For instance, an image with a heatmap representing the different types of filtrations.

5. Line 161 might be misleading and line 7 in the caption of table 1. The cardinality of the persistence intervals counts the k-dimensional cycles that appear and disappear or persist along the filtration. Intervals “at infinity” count the effective numbers of cycle in the data.

6. Line 445 449. Would add bullet points as for the other types of noise.

7. Table 5. It would be useful adding a legend to read the colors of the heatmap representing the accuracy.

8. Table 5. It is not clear what are the big squares in the first row for each filtration.

9. Table 5 caption. It is not clear what is the meaning of the last sentence. Does it refer to the big squares in the first row for each filtration?

10. Line 564. What are the assumptions of the filtration functions?

6. PLOS authors have the option to publish the peer review history of their article (what does this mean?). If published, this will include your full peer review and any attached files.

Reviewer #1: No

Reviewer #2: No

---

## [Author Response · Author response to Decision Letter 0]

13 Aug 2021

A detailed response to each of the reviewer and editor comments is attached as a response_to_reviewers_turkevs2021noise.pdf.

---

## [Editor Report · Decision Letter 1]

26 Aug 2021

Noise robustness of persistent homology on greyscale images, across filtrations and signatures

PONE-D-21-14381R1

Dear Dr. Turkes,

We’re pleased to inform you that your manuscript has been judged scientifically suitable for publication and will be formally accepted for publication once it meets all outstanding technical requirements.

Kind regards,

Giovanni Petri, Ph.D.

Academic Editor

PLOS ONE

Additional Editor Comments (optional):

I would recommend a thorough check before production as I spotted a few leftover typos in the revised version. Apart from that, the manuscript seems ready for publication.  
---

## [Editor Report · Acceptance letter]

13 Sep 2021

PONE-D-21-14381R1 

Noise robustness of persistent homology on greyscale images, across filtrations and signatures 

Dear Dr. Turkeš:

I'm pleased to inform you that your manuscript has been deemed suitable for publication in PLOS ONE. Congratulations! Your manuscript is now with our production department. 

Kind regards, 

on behalf of

Dr. Giovanni Petri 

Academic Editor

PLOS ONE